# A Review of Quantitative Systems Pharmacology Models of the Coagulation Cascade: Opportunities for Improved Usability

**DOI:** 10.3390/pharmaceutics15030918

**Published:** 2023-03-11

**Authors:** Douglas Chung, Suruchi Bakshi, Piet H. van der Graaf

**Affiliations:** 1Quantitative Systems Pharmacology, Certara UK Limited, Sheffield S1 2BJ, UK; 2Division of Systems Pharmacology and Pharmacy, LACDR, University of Leiden, 2300 RA Leiden, The Netherlands

**Keywords:** quantitative systems pharmacology, coagulation cascade, model reusability, mechanistic hypotheses exploration

## Abstract

Despite the numerous therapeutic options to treat bleeding or thrombosis, a comprehensive quantitative mechanistic understanding of the effects of these and potential novel therapies is lacking. Recently, the quality of quantitative systems pharmacology (QSP) models of the coagulation cascade has improved, simulating the interactions between proteases, cofactors, regulators, fibrin, and therapeutic responses under different clinical scenarios. We aim to review the literature on QSP models to assess the unique capabilities and reusability of these models. We systematically searched the literature and BioModels database reviewing systems biology (SB) and QSP models. The purpose and scope of most of these models are redundant with only two SB models serving as the basis for QSP models. Primarily three QSP models have a comprehensive scope and are systematically linked between SB and more recent QSP models. The biological scope of recent QSP models has expanded to enable simulations of previously unexplainable clotting events and the drug effects for treating bleeding or thrombosis. Overall, the field of coagulation appears to suffer from unclear connections between models and irreproducible code as previously reported. The reusability of future QSP models can improve by adopting model equations from validated QSP models, clearly documenting the purpose and modifications, and sharing reproducible code. The capabilities of future QSP models can improve from more rigorous validation by capturing a broader range of responses to therapies from individual patient measurements and integrating blood flow and platelet dynamics to closely represent in vivo bleeding or thrombosis risk.

## 1. Introduction

Dysregulation of hemostasis such as bleeding or thrombosis can be caused by acquired coagulation disorders (e.g., hemophilia), surgery/trauma, or chronic inflammatory conditions (e.g., atherosclerosis). Despite the numerous therapeutic options to stop bleeding or prevent blood clot formation, a mechanistic understanding of the effect of these therapies is not yet fully known but we have made great progress through mathematical models. These mathematical models have been developed to better understand the fundamental kinetics of the coagulation cascade and the mechanism of action of treatments for bleeding and thrombosis. Most of these models can be categorized as either systems biology (SB) or quantitative systems pharmacology (QSP) models. The BioModels database [1] provides the model code for both SB and QSP models of the coagulation cascade. Primarily, two SB models [2,3] and three QSP models [4,5,6] serve as the basis from which other published models have adapted to address specific research or clinical questions. Unfortunately, the shared components between these models are unclear and not all code can reproduce published results as previously reported [7,8,9].

Outside of the models available on the BioModels database, mathematical models have attempted to investigate the fundamental processes of hemostasis which include not only the coagulation cascade [4,5,6] but also vasospasm [10], platelet aggregation [11], activation [5,12], fibrin clot formation [4,5,6,12,13], and fibrinolysis [6,12,13] have been investigated using mathematical models but there is still a need for an integrated model to investigate bleeding or thrombosis risk for various patient types and therapeutic compounds. The promise of QSP models is that the collective knowledge of the known kinetics of the coagulation cascade and observed clinical phenomena is captured in the model code (i.e., equations, initial conditions, rate constants) and therefore is available for further research. By having reproducible and reusable models, this knowledge will not be lost and will only grow as existing models are adapted to explain increasingly more complex observations of hemostasis in patients. The focus of this paper is to review the literature on QSP models of the coagulation cascade to assess the unique reusability and capabilities of these models.

## 2. Models Overview

Both SB and QSP models are typically composed of a system of ordinary differential equations (ODE) representing the kinetics that describe the time courses of coagulation factors initiated by either the intrinsic or extrinsic pathways. SB models capture the nonlinear kinetics, autocatalytic feedback, and high sensitivity to initial conditions which are features of coagulation. Often, the action of calcium ions within membrane-bound enzyme complexes (e.g., prothrombinase) and phospholipid membrane-binding sites are implicitly included in the rate constants defining the kinetics [2,3]. Models typically assume an expected amount of activated/exposed platelet surfaces and availability of phospholipids by adjusting select equations of coagulation factor activation.

However, unlike SB models, QSP models are designed to interrogate the interaction of a single or group of proteases, cofactors, or regulators and its impact on thrombin formation and/or fibrin activation following interventions such as prothrombotic factors [6,14,15] or anticoagulants such as in clinical scenarios [4,16,17,18,19]. Comprehensive QSP models usually implement a portfolio of drug action mechanisms based on their physicochemical properties and pharmacokinetic profiles to better predict the time to blood clot formation based on both the drug and coagulation kinetics [19]. The field of coagulation has uncovered novel interactions between coagulation factors and recent QSP models have represented components related to the vitamin K cycle, antithrombin-III (AT-III), and tissue factor pathway inhibitor (TFPI) [6]. These newer areas of biology have improved the prediction in therapeutic response to treatments such as warfarin, vitamin K, and heparins by accounting for natural regulatory feedbacks that delay coagulation time.

Most QSP model simulations accurately predict the coagulation times for the standard in vitro coagulation tests measured in the clinic, PT and aPTT [4,5,6,20]. Because QSP models can simulate the protocols for the PT and aPTT tests, this allows a comparison between simulation data and clinical studies to aid in model qualification, the comparison between alternate therapies within the same model, and an assessment of the model performance by comparing the clotting time of the same therapy between different models. Models are validated by comparing the time required to reach threshold levels of thrombin to the time required to form clots in human whole-blood in vitro tests. Additional model outputs include the time courses for serine proteases such as activated factor X (FXa) or thrombin. Both proteases are part of the common pathway and are commonly used as markers of clot formation. In most models, the thrombin generation time course is simulated to compute quantitative parameters to help characterize the three phases of coagulation: initiation, propagation, and termination phases. These quantitative parameters include variations of the clotting time, thrombin peak time, thrombin peak height, maximum thrombin generation rate, and endogenous thrombin potential (Figure 1). The ability of therapeutic agents to normalize these parameters, related to thrombin generation, can be compared with conventional coagulation tests used to assess bleeding risk [21].

Since QSP models can accurately predict the clotting time and thrombin generation of in vitro experiments, they can be used to estimate clinical endpoints such as the international normalized ratio (INR) which is the standard measurement used to monitor patients on anticoagulants [4]. As QSP models become more precise in simulating the kinetics of the coagulation cascade, more recent models have taken steps to simulate in vivo coagulation. A few exploratory models have begun to examine the effects of blood flow on in vivo coagulation, accounting for the exchange of proteins between a developing clot and the fresh blood pool of coagulation factors [11,22].

Models can also directly compare the efficacy of multiple treatments for different clinical scenarios [4]. Bleeding or thrombotic disorders can be simulated by QSP models to understand the activity of serine proteases, cofactors, regulators, and fibrin. More complicated scenarios including congenital disorders (e.g., hemophilia) or treatment-induced bleeding following anticoagulants (e.g., warfarin, heparin) can also be explored by recreating these conditions in QSP models [23].

Mathematical models of the coagulation cascade have advanced from SB models investigating hypothetical interactions between coagulation factors to QSP models predicting therapeutic responses for different clinical scenarios. The current portfolio of available QSP models in the BioModels database [1] contains our most advanced understanding of the coagulation cascade and can accurately predict common in vitro clinical measurements. However, to facilitate widespread adoption, the capabilities and assumptions of these models need to be more clearly stated and the reusability of the model code would benefit from standard terminology and reproduction of published results.

## 3. Reusability

We reviewed over 80 SB and QSP models published over the past 32 years [2,3,4,5,6,11,12,13,14,15,16,17,18,19,21,22,23,24,25,26,27,28,29,30,31,32,33,34,35,36,37,38,39,40,41,42,43,44,45,46,47,48,49,50,51,52,53,54,55,56,57,58,59,60,61,62,63,64,65,66,67,68,69,70,71,72,73,74,75,76,77,78,79,80,81,82,83,84,85,86]. Most of the SB models are redundant with mainly two seminal models [2,3] serving as the basis for subsequent QSP models. Of the QSP models, only three models serve as a bridge connecting the seminal SB models with more recent QSP models incorporating the proteases, cofactors, regulators, and fibrin of the coagulation cascade comprehensively, and are highly cited [4,5,6].

Mapping which QSP models have borrowed from previously published models shows a long history of connections (Figure 2). The figure illustrates the relevance and importance of the reuse of QSP models. SB and QSP models of the coagulation cascade can be linked to each other by the reuse of coagulation kinetics and the purpose of model development which includes the simulation of bleeding and thrombosis scenarios and the effects of therapeutic molecules. Most links between models are not explicitly stated in the reference but can be identified in the model ODE’s and shared rate constants for corresponding reactions. As the field has matured, additional regulatory factors have begun to appear in more recent QSP models while maintaining the lineage from the seminal SB models. Moreover, the increased accessibility of coagulation test kits has increasingly validated the thrombin generation and coagulation factor dynamics of more recent models. Finally, the clinical relevance of QSP models has increasingly grown as in vivo coagulation endpoints have been successfully matched to patient data of bleeding and thrombosis scenarios.

The purpose of a significant percentage of QSP models was to investigate bleeding complications. Adams et al., 2003 [16] combined a highly cited model of the common pathway of the coagulation cascade [2] with a model of tissue factor (TF) initiation of thrombin generation [24] to assess the impact of thrombin inhibition. The model featured additional reactions describing the binding of each of the thrombin inhibitors TFPI and AT-III to thrombin and meizothrombin. The Shibeko et al., 2012 [25] model investigated the supraphysiological dosing requirement of recombinant activated factor VII (rFVIIa) for effective bleeding control. It consisted of a minimal set of reactions between a limited set of coagulation factors demonstrating how it is not necessary to fully adapt previously published models. Conversely, the ability of rFVIIa to modulate thrombin production was explored using a TF-initiated model [14] based on previously published models of thrombin generation [2,26]. An example of a model investigating the effects of clotting factor supplementation following hemorrhaging or blood dilution, the Mitrophanov et al., 2016 [15] and Govindarajan et al., 2016 [12] models simulated thrombin generation in various dilution and supplementation scenarios based on previous models [2,25,26].

Another related purpose for QSP models was to investigate clotting complications or thrombosis. The Orfeo et al., 2010 model [17] assessed direct FXa inhibitors, a common type of anticoagulant, by simulating thrombin generation initiated by either TF or blood resupply (i.e., transfusions). This model was based extensively on published models of the extrinsic pathway [2,24,27,28]. One of the earliest examples of QSP models investigating congenital effects on thrombosis, the Brummel-Ziedins et al., 2012 [29] model explored one of the most prevalent thrombophilic risk factors, protein C (PC) mutations, by combining a published model of TF-initiated thrombin generation [2], an empirically validated model of the PC pathway [30], and thrombomodulin dynamics to generate individual TG profiles based on initial concentrations of proteases, cofactors, and regulators.

Most of these QSP models extensively modified a previously published model without detailed documentation on how parts of the biological scope, rate constants, and assumptions were adapted. In general, the TF-mediated extrinsic pathway [2] and the factor XII (FXII)-mediated intrinsic pathway [3] in most QSP models are taken from the two seminal SB models as previously mentioned. Of the QSP models, the equations and parameters from a few references have been widely adopted in subsequent modeling studies which we term the “legacy” models. The three legacy QSP models incorporated the known proteases, cofactors, regulators, and fibrin comprehensively and are the most extensively validated for accurate responses to therapeutic drug effects compared to the remainder of QSP models.

The three legacy models include the Burghaus et al., 2011 model [4], the Chatterjee et al., 2010 model [5], and the Wajima et al., 2009 model [6]. The Chatterjee model expanded the reaction network of the extrinsic pathway [2] with an empirically derived PC pathway, alternate activation reactions of factor VII (FVII) and factor XI (FXI), FXII-mediated intrinsic pathway, irreversible thrombin entrapment with fibrin, and consideration of time-varying platelet activation (Figure 3). The Wajima model included the extrinsic and intrinsic pathways, and the effects of warfarin on the vitamin K-related factors, and was validated by comparing simulations of aPTT, and PT with experimental results following warfarin and heparins (Figure 4). The Burghaus model adapted the seminal SB models [2,3] for in vivo conditions and predicted the response to commonly used anticoagulant therapies (Figure 5). Notably, the PC/protein S (PS) system and the coagulation factor adsorption reactions to lipids were adapted from published SB models [31,32] with additional reactions reflecting potential inactivation paths. These legacy models provide a foundation for future QSP research by ensuring that the coagulation kinetics are captured comprehensively and are validated for investigating the treatment of bleeding with prothrombotic factors and thrombosis with anticoagulants. Although evidence of their potential reusability has already been seen in recently published QSP models [18,19,33], the model code is not reproducible which hinders the adoption and further advancement of QSP models.

The code for these models is only available on BioModels; however, they are not completely reproducible. For example, although the Burghaus model code can be accessed in the BioModels database, the available code is unable to recreate a key simulation of the thrombin generation curve PT test as published (Figure 6). Because the results are not reproducible, future researchers are hindered by having to estimate appropriate values for the rate constants of the coagulation kinetics without the full context of the prior model which may include implicit assumptions within the model equations. This traces back to identifiability issues and not following standard practices [8]. We recommend that references share a graphical schematic of the most significant elements of the model compartments, species, reactions, etc. Model equations and parameters need to be clearly provided with relevant boundary and initial conditions including a statement on the algorithm and its settings used to solve the equations. Most importantly, the model code and files generated to build and solve the models need to be made available along with any pseudo code for recreating model simulation results generated in the reference.

Depending on the purpose of the model, it may be more practical to simplify an existing model to only include specific mechanisms or therapies rather than fully adopt their full complexity. Model simplification focuses on only the necessary mechanisms in the coagulation cascade to facilitate modeling and simulation workflows and sometimes reveals the natural redundancies in the network of coagulation reactions. For example, the Zhou model [19] simplified the Wajima model [6] by removing components related to the vitamin K and warfarin effects to focus on the effects of FXa inhibitors. In another example, a reduced 5-state model [34] was derived from the original 62-state Wajima model [6] to simulate fibrinogen recovery following snake envenomation. The reduced model was able to explain the range of fibrinogen response in snake bite victims (n = 73) showing the decline and recovery of fibrinogen concentrations following brown snake envenomation. Furthermore, all the states and 9 out of the 11 total parameters in the simplified model were fully identifiable. Similarly, a reduced order model [18] consisting of 22 ODEs was based on the Chatterjee model [5] which included the simplified platelet activation. Model simplification can be insightful in revealing the sensitivities and potential redundancies in the coagulation cascade. In the Panteleev et al., 2010 [35] model, a previously published SB model was decomposed into a set of physiologically relevant subtasks (i.e., clotting threshold, triggering, control by blood flow velocity, spatial propagation, and localization) revealing the modular nature of the biochemistry of the coagulation cascade and the interconnections between subtasks. In the Shibeko model [25], sensitivity analysis of FVII activation steps was performed to develop a simplified FXa generation model while maintaining a thrombin generation profile consistent with thrombin generation assays using FVII-deficient plasma samples (n = 8). As more automated approaches for model simplification are developed, this may be an increasingly valuable tool for reusing existing QSP models.

## 4. Capabilities

QSP models can explore measurable hypotheses of the coagulation cascade. For example, model simulations showed the simultaneous existence of TF-dependent and phospholipid-dependent rFVIIa-induced coagulation and found that each mechanism is independent [5,14] as supported by in vitro experiments [23]. The interaction between the key factors VIIa and Xa responsible for the initiation of coagulation was investigated in the Shibeko et al., 2010 [36] model where the inhibition of TF-VIIa by TFPI and VII activation by Xa combined to create a threshold-like response. The threshold is sensitive to blood flow due to the rapid removal of Xa resulting in rapid clotting at low shear rates and nearly no thrombus at higher shear rates. The Chatterjee model [5] made a similar assumption by implementing the conversion of IX and X mediated by VIIa independent of TF only at high concentrations and only in the presence of activated platelets. This effect was found to be kinetically significant when used above nanomolar concentrations. QSP models have provided evidence of the impact of prekallikrein on the activation of FXII via the intrinsic pathway [5,6,19]. In another example, QSP models were able to simulate the effects of PC, AT-III, and thrombomodulin (Tm) on thrombin generation as seen in whole-blood in vitro experiments [15]. The Chatterjee model [5] captured the experimentally observed 1000-fold increase in Xa levels following platelet activation and the addition of a minimal amount of exogenous Xa. Furthermore, the model suggested that XII activation is mediated by a first-order dependence on XII concentration and the auto-activation of XI on negatively charged surfaces. In the Burghaus model [4], the kinetics of protein binding on phospholipid vesicles were directly included allowing for the approximation of both the solubilized and phospholipid-mediated coagulation reactions. Moreover, the model estimated the typical amount of TF and FXIIa in vivo was estimated to be between 0.01 and 10 pM, representing typical situations such as exposure to subendothelial tissue or contact activation.

QSP models help understand the mechanistic cause of clinical phenomena associated with treatment for bleeding or thrombosis. For example, supraphysiological dosing of rFVIIa is often required to cease bleeding but the reason for such large doses is unknown. A QSP model revealed that a high rFVIIa dose amount is necessary to overcome zymogen inhibition by endogenous FVII, except in the absence of TF, confirming that the dose amount is sensitive to the interaction between TF, FVII, and rFVIIa [14]. Under specific conditions, rFVIIa caused a dose-dependent increase in thrombin generation in both the presence and the absence of FVII. This effect was not impacted by phospholipids as shown by zero change in the relative inhibitory effect of FVII, the ratio of peak thrombin in the presence of FVII to the absence of FVII, under varying phospholipid concentrations. QSP models can also explore bleeding due to congenital or treatment-induced conditions. Hemophilia A and B can be simulated using QSP models by virtually removing factor VIII (FVIII) or factor IX (FIX), respectively, as seen in several models [6,14,18]. Simulations showed that treatments using supraphysiological dosing of FVIIa (up to 20 nM) accelerated thrombin generation for FVIII/FIX-deficient blood, unlike normal blood where FVIIa only affects the propagation phase of thrombin generation [6,14]. In a prospective randomized trial, a high dosage of recombinant FVIIa successfully treated hemarthroses in hemophiliacs [87]. QSP models can offer hypotheses for unexpected thrombosis or bleeding. For instance, blood clot formation with no addition of TF was hypothesized to be triggered by trace amounts of FXIIa. The Chatterjee model [5] was able to reproduce the unexplainable, spontaneous clotting that can occur in some blood samples that have even been treated with an FXIIa inhibitor, corn trypsin inhibitor (CTI). The model was confirmed by experiments showing that the combination of anti-XI and anti-XII antibodies prevented such spontaneous clotting, demonstrating that a leak of FXIIa past saturating amounts of CTI is responsible for in vitro initiation without added TF. For trauma/hemorrhage patients, the delay in clotting typically following the administration of resuscitation fluids was explained by QSP models to be possibly caused by a dilution of coagulation factors [5,6,25,27]. In another scenario, bleeding occurred following the supplementation of depleted blood (less than 0.1% activity of prothrombin, FV, FVII, FVIII, FIX, or FX) with prothrombin complex concentrates, where simulations uncovered the conditions at which normal thrombin generation is unable to be restored [15]. In the same model, the treatment of plasma dilution with supplementation of coagulation factors was simulated using prior models [14]. The model showed good agreement with patient samples (n = 10, 3-fold dilution) and showed that only CCF-AT (composed of FII, FIX, FX, and AT-III) could restore normal thrombin generation in diluted plasma whereas rFVIIa (up to 40 nM added above baseline) and CCF-FVII (composed of FII, FVII, FIX, and FX) failed. For thrombosis patients, the ability of the anticoagulant rivaroxaban to delay clotting was explained by a QSP model whose simulations were consistent with the delayed clot prolongation times (2- to 3-fold clotting times) from a whole blood assay with a dose of 10 mg per day which is also consistent with the standard range of INR values used to guide anticoagulant therapy [17]. For thrombosis caused by envenomation, a population PKPD model based on the Wajima model [6] was built to investigate the treatment of brown snake envenomation which contains a potent prothrombin activator (Xa:Va) with an in silico antivenom administration [34]. The model matched profiles of fibrinogen concentration-time data from whole blood in vitro experiments.

QSP models can evaluate the mechanism of action of anticoagulants and directly compare the effects between them. QSP models have evaluated many anticoagulants including direct thrombin inhibitors: argatroban, (xi)melagatran, efegatran, and hirudin [4,16]; direct tenase (FXa) inhibitors: rivaroxaban [4,17,18,19], fondaparinux [17], apixaban [19]; and vitamin K inactivity due to warfarin (INR > 1.5) [17]. In the Orfeo et al., 2010 model [17], the interactions between tenase inhibitors: fondaparinux, rivaroxaban, and coagulation factors: AT-III, FXa, FIXa, FXa, Xa:Va, meizothrombin, and thrombin were evaluated by optimizing the correspondence between predicted and observed peak thrombin levels. The simulations suggested that the enhanced anti-FIXa activity of fondaparinux-AT-III may be critical to its success in acute settings in vivo. In the Burghaus model [4], the safety and efficacy of a portfolio of different classes of anticoagulant drugs were compared to help guide treatment selection. Notably, simulations predicted that the therapeutic window for rivaroxaban is robust due to significant efficacy achieved at C_trough_ concentrations and minimal reduction in peak thrombin at C_peak_ concentrations which is consistent with the approved total daily dose for rivaroxaban between 5 mg and 40 mg. In the same model, a comparative analysis between rivaroxaban and warfarin showed that the drug effect of rivaroxaban is dependent on the TF initiation level whereas the effects of warfarin are independent of TF initiation. In the Adams et al., 2003 model [16], low-molecular-weight heparins were evaluated in different patient types. Simulations showed that most direct thrombin inhibitors were unable to completely suppress thrombin generation after 40 min even at their highest concentrations (250 nM) whereas less than 50 nM of hirudin was predicted to ablate active thrombin even after 7 h in human whole blood. However, even 0.5% preactivated factor Va (FVa) was found to ablate the anticoagulant effect of hirudin even up to 250 nM. Since all the inhibitors strongly depend on preactivated FVa concentrations, the dosing of patients without prior knowledge of these circulating activated coagulation factors could be detrimental. The investigators concluded that dosing below a threshold level of circulating activated factors only delays active thrombin generation but dosing above the threshold nearly completely suppresses the production of thrombin due to the small quantities of thrombin required for the activation of cofactors V and VIII. In the Zhou et al., 2015 model [19], the plasma pharmacokinetics of widely used FXa inhibitors such as rivaroxaban, and apixaban were integrated with a mechanistic model of the coagulation cascade. Simulated profiles of the drug effect following multiple dosing closely matched the aPTT and the dose–response relationship with PT. Sensitivity analysis of the QSP model showed that the responses of FXa and fibrin were sensitive to the target binding kinetics of the direct FXa inhibitors which could guide the development of future therapies.

QSP models can also offer hypotheses for significant phenomena which have been observed in vivo. For example, thrombus formation after successful anticoagulant treatment could be caused by preactivated proteases from a previous thrombus (blood resupply) without the need for TF initiation. A model predicted that only rivaroxaban is effective at suppressing clotting due to both blood resupply and ongoing coagulation due to its higher reactivity towards the prothrombinase complex [17]. Simulations showed that rivaroxaban can effectively suppress new thrombin formation that derives from preformed thrombin while fondaparinux cannot. The inability of fondaparinux to fully suppress thrombin generation during blood resupply in simulations or experimental models contrasts with its ability to suppress TF-initiated coagulation, suggesting that simply increasing the dose may not be effective during an active thrombus. In the Brummel-Ziedins et al., 2012 model [29], simulations set to the plasma composition of individual patients with or without a PC mutation (homo/heterozygous familial PC deficiency) showed that carriers have greater thrombin generation than individuals that do not. Interestingly, women possessing the PC mutation have a significantly faster clot time compared to those who do not, but no such difference was found in males.

QSP models expand the capabilities of research in the field of coagulation (Table 1). By bridging the complex interactions between coagulation factors, therapies, and other mediators, clinical observations can be studied at a mechanistic level. With further development, QSP models will begin to predict the therapeutic window for existing therapies in untested clinical scenarios and patient types. The development of new classes of anticoagulants or prothrombotic factors would also benefit from in silico experiments to optimize dosing and patient selection.

## 5. Future Direction

The field of blood coagulation appears to suffer from the same issues of a lack of reproducibility from source code as previously reported for SB and QSP models in other therapeutic areas [7,8,9]. Later QSP models selectively reused and modified the components from previously published models to adapt for different purposes and for explaining a distinct set of experimental data, often without listing the changes or rationale. The unclear links between models and irreproducible code impede the application of these models for specific uses such as in drug development. The field would benefit from a concerted effort to minimize redundancies in the model equations, develop a standard for PT and aPTT simulation protocols including platelet effects, clearly link the model equations and parameters between models, and share reproducible code that matches the published simulations of the thrombin generation curve and clotting times from aPTT and PT.

Future QSP models would benefit from using the legacy models to ensure that the intrinsic, extrinsic, and common pathways of the coagulation cascade are comprehensively represented and that only the validated kinetics of the proteases, cofactors, regulators, fibrin, and effects of anticoagulants are propagated in future models. More recent models have begun to adopt these legacy models for investigating in vivo biomarkers [33], the effects of hemophilia on coagulation [18], and the effects of anticoagulants [19,34,37]. To facilitate their adoption, even more, further validation of legacy models would increase confidence in their predictions of clotting times, thrombin generation curves, and response to therapies. Because the kinetics of thrombin generation is sensitive to initial concentrations of coagulation factors, simulating profiles for individual patient whole-blood samples could be used to fine-tune these models to be able to explain a broader range of thrombin generation and clotting times for different blood compositions. The legacy models could be more robust by explaining more extreme scenarios of in vitro blood clotting such as the effects of very high doses of warfarin (INR > 9) with a physiological clotting time and thrombin level.

QSP models can predict the differences in individuals’ responses to anticoagulants to personalize dosing regimens including titration protocols of anticoagulants to achieve an optimal INR window. For bleeding events, the effect of prothrombotic factors is sensitive to an individual’s endogenous zymogens and other factors including blood dilution due to resuscitation fluids. QSP models could help tailor individualized dosing of these prothrombotic factors based on the patient’s blood composition and treatment history. However, these differences in hemostasis between individuals as seen in current clinical treatment protocols [29,30] require additional studies including in vitro testing and in vivo animal and human whole blood assays. The measurement data from these future studies could be analyzed using QSP models to help generate hypotheses for the observed variability in hemostasis.

It is worth noting that all the published QSP models focus on in vitro measurements such as the thrombin generation assay, PT, and aPTT. Very few models, if any, have modeled the in vivo thrombus formation. Even models incorporating blood flow, platelet dynamics, and fibrin generation are empirical with the aim of capturing in vitro clotting under blood flow such as 2-dimensional flow chambers with thrombogenic surfaces [12] or thromboelastography [38]. It is not clear how the model rate constants which capture the coagulation kinetics under in vitro experimentation directly translate to the highly complex in vivo situation. There has not yet been a comprehensive model to predict the in vivo risk of thrombosis or bleeding. This is because many other factors including vasospasm, platelet aggregation and activation, fibrin clot formation, and fibrinolysis all affect the initiation and progression of thrombus formation. A major step would be to recreate the observed effects of blood resupply and blood dilution in the legacy models to improve the relevancy of these models for in vivo situations. A longer-term goal would be to incorporate existing models of platelet dynamics, blood flow, and their effects on the coagulation cascade [22]. For example, the first spatial–temporal mathematical model of platelet aggregation and blood coagulation under flow simulated how the porous nature of a growing platelet mass allows inactivated platelets and zymogens to flow into and diffuse within a growing thrombus aiding in the sharp increase in thrombin and prothrombinase formation [11]. These concepts were expanded to include the effects of fibrin generation and their mechanical effects on coagulation kinetics and platelet activation [12]. Integrating the spatiotemporal effects of thrombus formation may eventually enable QSP models to simulate not only in vitro assays but in vivo thrombus formation.

QSP models have shown their predictive capability in the response to different treatments in human blood samples and have shown their ability to explain observed in vitro and in vivo phenomena at a mechanistic level. With further validation, these models could be important tools for designing and developing new classes of therapies. Eventually, QSP models may accurately predict the efficacy of anticoagulants and prothrombotic factors for in vivo clinical scenarios.

## Figures and Tables

**Figure 1 pharmaceutics-15-00918-f001:**
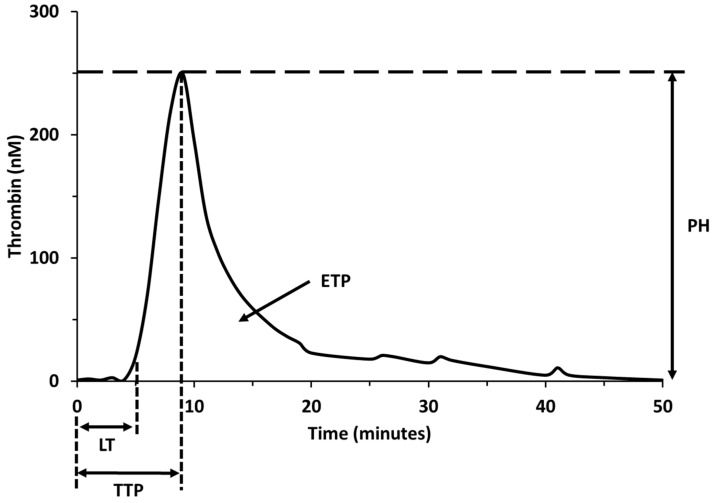
Quantitative parameters of thrombin generation. Lag time (LT); time to peak (TTP); thrombin peak height (PH); area under the thrombin curve or endogenous thrombin potential (ETP).

**Figure 2 pharmaceutics-15-00918-f002:**
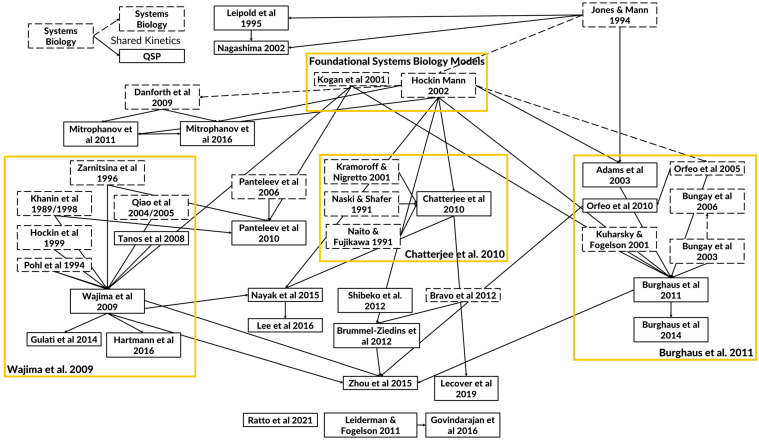
Diagram of key quantitative systems pharmacology (QSP) models of the coagulation cascade. The arrows depict how the whole or parts of the model (e.g., kinetics) have been adopted and modified between published models. The solid boxes indicate QSP models, and the dashed boxes indicate SB models. The solid arrows are used to link QSP models that share model equations and parameters from the original to the adopted model. The dashed arrows are used to link SB models. Models that are listed without any connections do not share significant common components. From over thirty years of mathematical models, there have emerged foundational systems biology models and three separate foci containing the legacy QSP models (shown in yellow boxes) with distinct applications [2,3,4,5,6,11,12,13,14,15,16,17,18,19,22,24,25,26,27,29,30,31,32,33,34,35,37,43,44,49,51,52,53,56,66,67,68,69,70,73,74,80,85,86].

**Figure 3 pharmaceutics-15-00918-f003:**
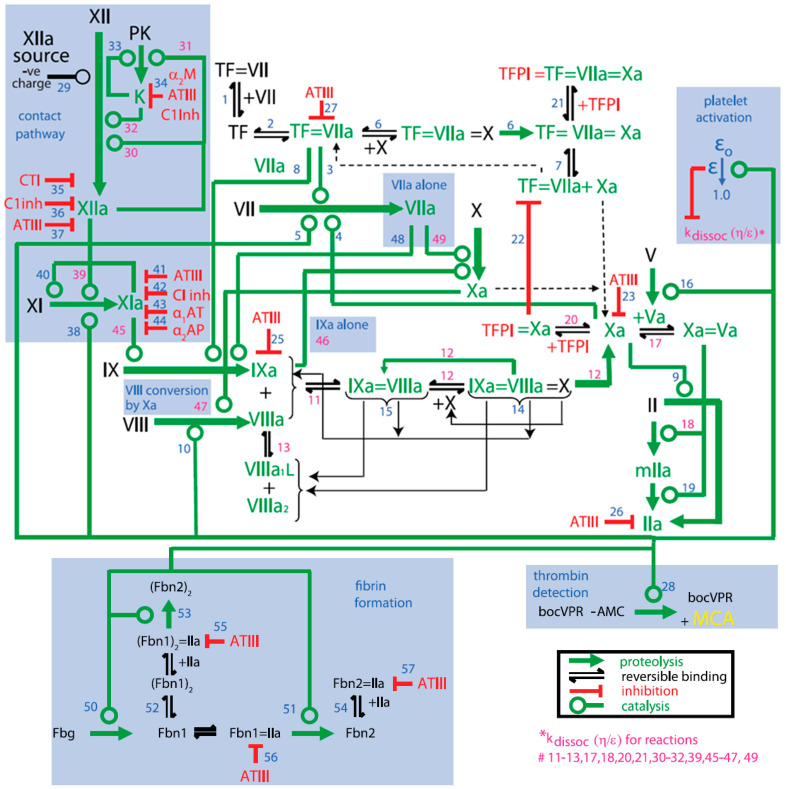
Schematic of the Platelet–Plasma dynamics from the Chatterjee model [5]. Blue highlighted portions represent additions to the Hockin–Mann model [2]. Alpha-1 antitrypsin (α1AT); alpha-2 antiplasmin (α2AP); alpha 2-macroglobulin (α2M); antithrombin-III (ATIII); C1-inhibitor (C1Inh); corn trypsin inhibitor (CTI); fibrin I (Fbn1); fibrin II (Fbn2); fibrinogen (Fbg); kallikrein (K); meizothrombin (mIIa); platelet activation factor (ε); prekallikrein (PK); tissue factor (TF); prothrombin (II); thrombin (IIa); tissue factor pathway inhibitor (TFPI). Blue numbers are the reaction steps in the model and pink numbers are those reactions whose dissociation rate constant is modified by the platelet activation factor (ε).

**Figure 4 pharmaceutics-15-00918-f004:**
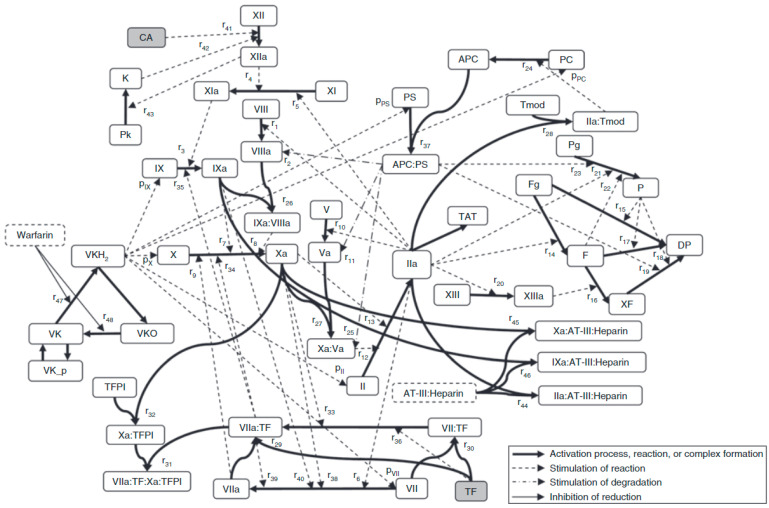
The scheme of the coagulation network model from the Wajima model [6]. The bold solid lines represent the activation process, complex formation, reduction, or oxidization; the broken lines represent stimulation of reaction or production; the dot-bar lines represent stimulation of degradation; the solid lines represent inhibition of reduction. Activated protein C (APC); antithrombin-III (AT-III); activator for the contact system (CA); degradation product (DP); fibrin (F); fibrinogen (Fg); prothrombin (II); thrombin (IIa); prekallikrein (Pk); kallikrein (K); plasmin (P); protein C (PC); plasminogen (Pg); protein S (PS); thrombin–antithrombin complex (TAT); tissue factor (TF); tissue factor pathway inhibitor (TFPI); thrombomodulin (Tmod); vitamin K (VK); vitamin K hydroquinone (VKH2); vitamin K epoxide (VKO); cross-linked fibrin (XF). See original publication [6] for more details on species and reactions.

**Figure 5 pharmaceutics-15-00918-f005:**
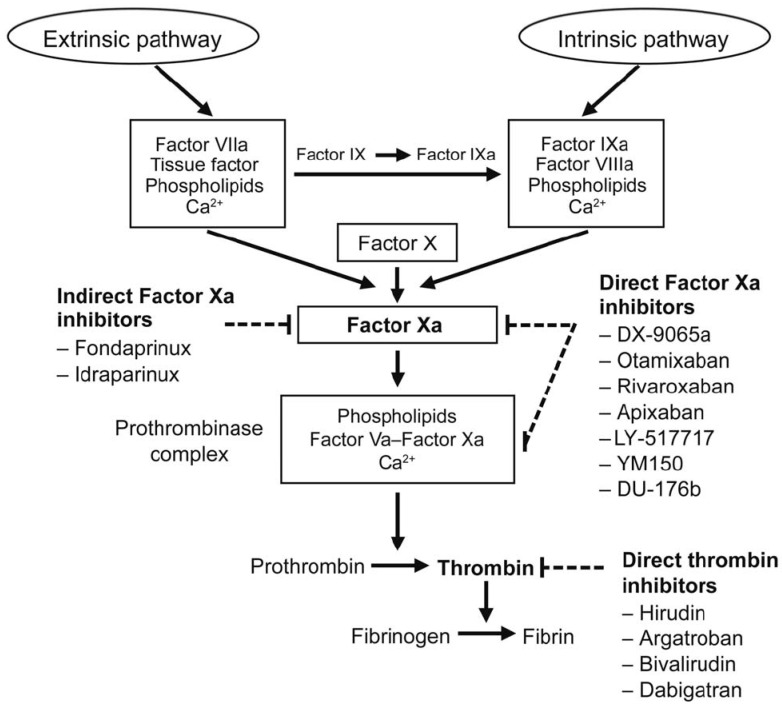
Targets for anticoagulant drugs in the coagulation pathway from the Burghaus model [4].

**Figure 6 pharmaceutics-15-00918-f006:**
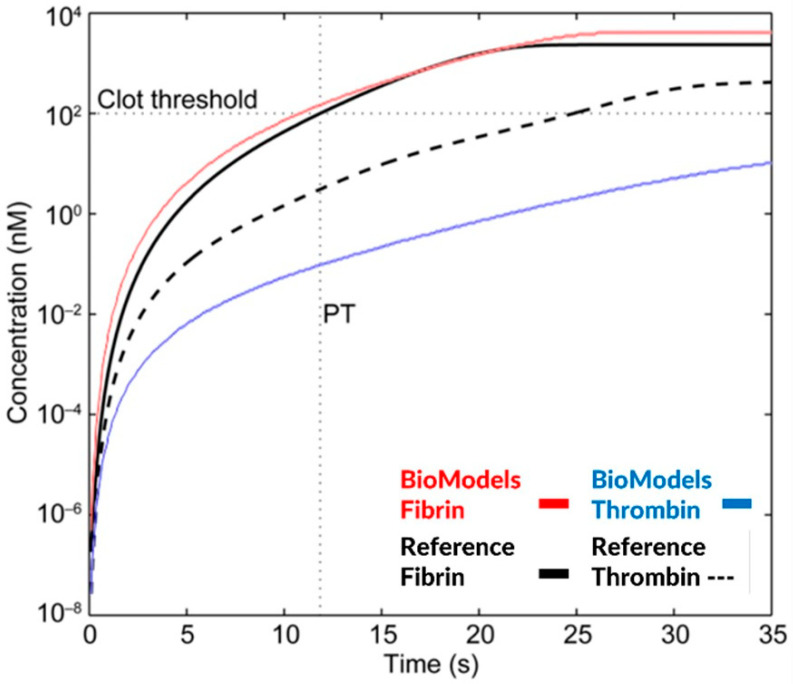
Simulation of a prothrombin time test (PT) based on thrombin and fibrin generation for the Burghaus model [4] and the related BioModels code repository [1]. Fibrin concentrations are similar; however, the thrombin generation curve is significantly different between the simulation results of the published and the publicly available model code.

**Table 1 pharmaceutics-15-00918-t001:** QSP models offer insights into the mechanisms of the coagulation cascade, in vivo coagulation, and treatment effects for thrombosis and bleeding.

QSP Model Capabilities	Description	References
Investigate the mechanism(s) of the coagulation cascade	The simultaneous existence of TF-dependent and phospholipid-dependent rFVIIa-induced coagulation where each mechanism is independent	[5,14,23]
	Inhibition of TF-VIIa by TFPI and VII activation by Xa combine to create a threshold-like response of thrombin generation	[5,36]
	Impact of prekallikrein on the activation of FXII via the intrinsic pathway	[5,6,19]
	Effects of PC, AT-III, and thrombomodulin ™ on thrombin generation (whole-blood in vitro experiments)	[15]
	1000-fold increase in Xa levels following platelet activation (whole-blood in vitro experiments)	[5]
	Auto-activation of XI on negatively charged surfaces	[5]
Increase understanding of in vivo coagulation	Estimated the typical amount of TF and FXIIa in vivo	[4]
	Hemophilia A and B can be simulated using QSP models	[6,14,18]
	PC mutation carriers have greater thrombin generation than individuals that do not	[29]
Increase understanding in the treatment of thrombosis, bleeding (hemostasis)	A high rFVIIa dose amount is necessary to overcome zymogen inhibition by endogenous FVII	[14]
	Treatments using supraphysiological dosing of FVIIa accelerate thrombin generation for FVIII/FIX-deficient blood	[6,14]
	Simulations uncovered the conditions at which normal thrombin generation is unable to be restored (bleeding) following the supplementation of depleted blood with prothrombin complex concentrates	[15]
	Delay in clotting typically following the administration of resuscitation fluids caused by a dilution of coagulation factors	[5,6,17,27]
	Predict the delayed clot prolongation times of the anticoagulant rivaroxaban in patient whole blood	[17]
	Investigate the treatment of brown snake envenomation	[34]
	Evaluate direct thrombin inhibitors	[4,16,17,18,19]
	The drug effect of rivaroxaban is dependent on the TF initiation level whereas the effects of warfarin are independent of TF initiation	[4]
	Direct thrombin inhibitors strongly depend on preactivated FVa concentrations (ablate thrombin generation)	[16]
	Response of FXa and fibrin were sensitive to the target binding kinetics of direct FXa inhibitors	[19]
	Rivaroxaban is effective at suppressing clotting due to both blood resupply and ongoing coagulation due to its higher reactivity towards the prothrombinase complex	[17]

## Data Availability

The data presented in this study are openly available in [BioModels] at [https://doi.org/10.1093/NAR/GKX1023], reference number [1].

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
