# Peer review of "A Review of Quantitative Systems Pharmacology Models of the Coagulation Cascade: Opportunities for Improved Usability"

_pharmaceutics, 2023, doi:10.3390/pharmaceutics15030918_

Round 1

Reviewer 1 Report

I read the manuscript of Chung and colleagues with great interest. The review provides an elaborate overview of the quantitative pharmacology models of the coagulation cascade. The topic is of interest for a large variety of researchers in the fields of thrombosis and hemostasis and systems biology. The manuscript is well-written and easy to read. I have some remarks/suggestions that might help the authors to improve the manuscript.

-        Could the authors elaborate on the role that pharmacology system models (could) play specifically in the field of thrombosis and hemostasis?

-        In sections 1 and 2 (introduction and models overview) references are missing, such as (but not exclusively) to seminal papers for the functioning of the coagulation cascade, the PT and aPTT tests, and anticoagulant treatment. Also, in line 43 and line 85 references are included (10 and 11, resp.) that are not corresponding to the references in the manuscript. Of note, these references are the first ones that are included in the manuscript and I do not believe that they refer to reference 10 and 11 of the current manuscript. Also during the course of the rest of the manuscript only references to SB and QSP models are included, and I believe that the manuscript should be critically revised to include additional references to add more context to the review.

-        Section 4 (capabilities): This section includes a lot of text without figure(s), and I believe that including (sub)headers in this section will improve the readability. Perhaps also a table could be included, in which studies/references are grouped, including their most relevant results.

-        Line 114: It is stated that 82 models are published, while 83 references (2-84) are included.

-        Line 174: Could the authors specify with what they mean exactly with a “legacy model”? I do not believe that this term is commonly used and I believe that it would be helpful for the reader to specify what the authors exactly mean with this term.

-        Lines 212-224: The problem that is posed here is clear, but how would the authors like to see this problem getting resolved?

-        Line 272: “Auto-activation of XI on negatively charged surfaces”. I believe that this should be auto-activation of XII.

-        Lines 291-294: It could be of interest to reference to studies that demonstrate the feasibility of this approach (FVIIa treatment) to treat hemophilia patients.

-        Line 302: What do the authors mean with “depleted blood”? It is not clear for which factor(s) the blood is depleted.

-        Line 370: “with or without a PC mutation”. Could the authors specify which PC mutation?

Author Response

Thank you for your helpful suggestions regarding the manuscript. By addressing the feedback regarding citing references, providing more detail for key ideas, and proofreading, we feel the manuscript will be of greater interest for researchers in the field of thrombosis/hemostasis and modeling. We hope that the following edits to the manuscript address the referee’s remarks/suggestions.

Please see the point-by-point response to your comments in the attached document. Thank you.

Kind regards,

Doug Chung

Reviewer 2 Report

This review concerning the quantitative systems pharmacology models of the coagulation cascade is a nicely presented and well written overview of the current state of the art, which might give impetus to further developments. I have only minor points to be addressed in the revision process. In the opinion of the reviewer, the manuscript can be published after minor revision.

Comments:

1. Introduction: More references need to be cited in this section.

2. Line 35: change to „blood clot formation“

3. Models Overview: Again, where are references cited? One reference only given at the end of the page is not enough for three paragraphs.

4. Line 73: Delete comma between „PT“ and „and“

5. Line 73: Utilizing the PT, aPTT allows…? I don't understand this, please re-write.

6. Line 114-115: Change to „Most of the SB models are redundant…“

7. Figure 2 description: Add the meaning of the yellow boxes or add the color in your descriptions. In addition, a dot is missing at the end of the description.

8. Line 173: Write „two“ instead of „2“

9. Figure 3: What are the numbers? Why different colors of the numbers? Dissociation and association constants? Not clear enough from the figure/figure legend.

10. Resolution of figures is too low, especially in case of figure 4.

11. Are there models that consider time and location dependency of the effects? Or models that try to include this? Please elaborate more on this aspect.

12. Concentration changes? (e.g. acute phase proteins of the blood coagulation system)

13. Again, references are missing in many paragraphs of this manuscript.

14. Line 276: Numbers superscript or provide calculations in "pM"

15. Line 336: Add space between number and mg

16. Chapter 4. Capabilities is difficult to read or to understand without a figure. Authors are encouraged to provide a figure here.

Author Response

Thank you for your helpful suggestions regarding the manuscript. By addressing the feedback regarding citing references, providing more detail for key ideas, and proofreading, we feel the manuscript will be of greater interest for researchers in the field of thrombosis/hemostasis and modeling. We hope that the following edits to the manuscript address the referee’s remarks/suggestions.

Please see the point-by-point response to your comments in the attached word doc. Thank you!

Kind regards,

Doug Chung

Reviewer 3 Report

Manuscript: pharmaceutics-2198169

Title: A Review of Quantitative Systems Pharmacology Models of the Coagulation Cascade: Opportunities for Improved Usability

Authors: Chung et al.

The manuscript raises many doubts and reservations. It is based on declarations and speculations. It also does not bring any new values in the fired of medical sciences. For this reason, in the reviewer’s opinion, the manuscript should be rejected.

List of some major errors:

1. Authors’ affiliation. In the case of Certara UK Limited, authors should provide more detailed information, such as the name of the department or laboratory. Also, in the case of the University of Leiden, authors should provide at least the name of faulty and department. Affiliation can only be used in the case of current employment in the listed units.

2. In the abstract and introduction, the authors stated that “despite the numerous therapeutic options to treat bleeding or thrombosis, a comprehensive quantitative mechanistic understanding of the effects of these and potential novel therapies is lacking”. They than present the current models of quantitative system pharmacology that have so far not contributed to the suggested understanding. These models are based on schemes that have been present in medical textbooks as well review articles on coagulation for years. The differences between concepts of these models (Figures 3-4) come down only the number of mentioned elements affecting the dynamics of coagulation and forms of graphical presentation. These methods have not reached the stage that allow their practical use in diagnosis or therapy. They present only attempts to create an idea of future methods and the set of elements that should be taken into account during creation of functional model. Thus, these activities have not, so far, gone beyond the stage of speculation and declarations. These models have not been verified in animal models or clinical study.

3. On the other hand, lack of animal or clinical verification of the quantitative systems pharmacology (QSP) models should be treated as a sign of common sense of researchers, because, as authors rightly noted, QSP models can only stimulate known interactions (line 45). The current knowledge in the field of hemostasis is still not complete, and there are large individual differences in the reaction of hemostatic system. Therefore, research still needs to be conducted in animal models or in clinical settings. Diagnosis must be based on the current biochemical tests, as well as therapy must be adapted to the current clinical situation during treatment.

Author Response

Thank you for your questions and remarks regarding the manuscript. We hope that the following edits to the manuscript and responses address the referee’s remarks/suggestions.

Please see the point-by-point response to your comments in the attached word doc. Thank you!

Kind regards,

Doug Chung

Round 2

Reviewer 3 Report

Manuscript: pharmaceutics-2198169 the second review

Title: A Review of Quantitative Systems Pharmacology Models of the Coagulation Cascade: Opportunities for Improved Usability

Authors: Chung et al. 

In a previous review, the reviewer suggested rejection of the manuscript. However, the other reviewers and the editor felt that the final decision on the manuscript should be made after it has bee revised. Taking this into account, below are the condition that must be met by the manuscript in order to make decision about its publication.

1.      A scientific article must be objective. It must not give the impression of and advertising offer. The current form of the abstract presents only super-optimistic assumption about possible role and usefulness of quantitative systems pharmacology models of the coagulation cascade. On the other hand, the abstract of the reviewer article should also include a detailed description of current achievement in this area, their shortcomings, and limitations, as the lack of functional integrated models covering the entire coagulation system, which would be positively verified in experimental or clinical trails.

2.      When presenting references to support their pint of view, the authors should provide more details about articles cited. For example (lines 90-91), the authors state that: “Most QSP model simulations accurately predict the coagulation times for the standard in vitro coagulation tests measured in the clinic, PT and aPTT [20]”. Firstly, if the authors state that “most QSP model simulations” then why do they present only one paper. Are there not more papers addressing this problem? Secondly, the authors should state in detail that that paper deals with a fragmentary aspect of the coagulation system, namely the prognosis of the effects of factor Xa inhibition in healthy individuals. The number of observations should be also reported and whether all real values observed in healthy subjects were within the range of predicted values. In the same way, the data presented in the other references should be presented in detail.

3.       At the end of the manuscript, the authors should also write that the current knowledge in the field of hemostasis is still not complete, and there are large individual differences in the reaction of hemostatic system. Therefore, research still needs to be conducted in animal models or in clinical settings. Diagnosis must be based on the current biochemical tests, as well as therapy must be adapted to the current clinical situation during treatment, which limits the usefulness of quantitative systems pharmacology models of the coagulation cascade in clinical settings now and in the near future.

Author Response

  1. A scientific article must be objective. It must not give the impression of and advertising offer. The current form of the abstract presents only super-optimistic assumption about possible role and usefulness of quantitative systems pharmacology models of the coagulation cascade. On the other hand, the abstract of the reviewer article should also include a detailed description of current achievement in this area, their shortcomings, and limitations, as the lack of functional integrated models covering the entire coagulation system, which would be positively verified in experimental or clinical trails.

Thank you for the comment. We agree with the reviewer that the article provides a promising outlook for the future of QSP models, hence the need for reusable and reproducible model code and simulations as one of the main points of this article. However, we are realistic about the current capabilities and limitations of QSP models as stated in the following examples.

Lines 427-429. “With further development, QSP models will begin to predict the therapeutic window for existing therapies in untested clinical scenarios and patient types.”

Lines 437-443. “The field of blood coagulation appears to suffer from the same issues of a lack of reproducibility from source code as previously reported for SB and QSP models in other therapeutic areas [7,8,9]. Later QSP models selectively reused and modified the components from previously published models to adapt for different purposes and for explaining a distinct set of experimental data, often without listing the changes or rationale. The unclear links between models and irreproducible code impede the application of these models for specific uses such as in drug development.”

Lines 482-487. “There has not yet been a comprehensive model to predict the in vivo risk of thrombosis or bleeding. This is because many other factors including vasospasm, platelet aggregation and activation, fibrin clot formation, and fibrinolysis all affect the initiation and progression of thrombus formation. A major step would be to recreate the observed effects of blood resupply and blood dilution in the legacy models to improve the relevancy of these models for in vivo situations.”

  1. When presenting references to support their pint of view, the authors should provide more details about articles cited. For example (lines 90-91), the authors state that: “Most QSP model simulations accurately predict the coagulation times for the standard in vitro coagulation tests measured in the clinic, PT and aPTT [20]”. Firstly, if the authors state that “most QSP model simulations” then why do they present only one paper. Are there not more papers addressing this problem? Secondly, the authors should state in detail that that paper deals with a fragmentary aspect of the coagulation system, namely the prognosis of the effects of factor Xa inhibition in healthy individuals. The number of observations should be also reported and whether all real values observed in healthy subjects were within the range of predicted values. In the same way, the data presented in the other references should be presented in detail.

Thank you for your comment. I have added references and more details on the type of patient data and how it was used in the QSP models.

Lines 90-91. "Most QSP model simulations accurately predict the coagulation times for the standard in vitro coagulation tests measured in the clinic, PT and aPTT [4,5,6,20]. "

Lines 277-285. “In another example, a reduced 5-state model [34] was derived from the original 62-state Wajima model [6] to simulate fibrinogen recovery following snake envenomation. The reduced model was able to explain the range of fibrinogen response in snake bite victims (n=73) showing the decline and recovery of fibrinogen concentrations following brown snake envenomation. Furthermore, all the states and 9 out of the 11 total parameters in the simplified model were fully identifiable.”

Lines 292-295. “In the Shibeko model [25], sensitivity analysis of FVII activation steps was performed to develop a simplified FXa generation model while maintaining a thrombin generation profile consistent with thrombin generation assays using FVII-deficient plasma samples (n=8).”

Lines 354-363. “In another scenario, bleeding occurred following the supplementation of depleted blood (less than 0.1% activity of prothrombin, FV, FVII, FVIII, FIX, or FX) with prothrombin complex concentrates, where simulations uncovered the conditions at which normal thrombin generation is unable to be restored [15]. In the same model, the treatment of plasma dilution with supplementation of coagulation factors was simulated using prior models [14]. The model showed good agreement with patient samples (n=10, 3-fold dilution) and showed that only CCF-AT (composed of FII, FIX, FX, and AT-III) could restore normal thrombin generation in diluted plasma whereas rFVIIa (up to 40 nM added above baseline) and CCF-FVII (composed of FII, FVII, FIX, and FX) failed.”

  1. At the end of the manuscript, the authors should also write that the current knowledge in the field of hemostasis is still not complete, and there are large individual differences in the reaction of hemostatic system. Therefore, research still needs to be conducted in animal models or in clinical settings. Diagnosis must be based on the current biochemical tests, as well as therapy must be adapted to the current clinical situation during treatment, which limits the usefulness of quantitative systems pharmacology models of the coagulation cascade in clinical settings now and in the near future.

Thank you for this comment. We agree that there are large individual differences in hemostasis that cannot be simply attributed to differences in coagulation factor concentrations. Also, as you stated, the model predictions are limited by the quality of blood sample measurements and current clinical treatment protocols. Please see the following addition to the paper.

Lines 470-474. “However, these differences in hemostasis between individuals as seen in current clinical treatment protocols [29,30] require additional studies including in vitro testing and in vivo animal and human whole blood assays. The measurement data from these future studies could be analyzed using QSP models to help generate hypotheses for the observed variability in hemostasis.”